# Technology for Our Future? Exploring the Duty to Report and Processes of Subjectification Relating to Digitalized Suicide Prevention

**Tineke Broer**

Tilburg Institute for Law, Technology, and Society, Department of Law, Technology, Markets, and Society, Tilburg University, Montesquieu Building, 7th floor, 5037 DB Tilburg, The Netherlands; T.Broer_1@uvt.nl

**Abstract:** Digital and networking technologies are increasingly used to predict who is at risk of attempting suicide. Such digitalized suicide prevention within and beyond mental health care raises ethical, social and legal issues for a range of actors involved. Here, I will draw on key literature to explore what issues (might) arise in relation to digitalized suicide prevention practices. I will start by reviewing some of the initiatives that are already implemented, and address some of the issues associated with these and with potential future initiatives. Rather than addressing the breadth of issues, however, I will then zoom in on two key issues: first, the duty of care and the duty to report, and how these two legal and professional standards may change within and through digitalized suicide prevention; and secondly a more philosophical exploration of how digitalized suicide prevention may alter human subjectivity. To end with the by now famous adagio, digitalized suicide prevention is neither good nor bad, nor is it neutral, and I will argue that we need sustained academic and social conversation about who can and should be involved in digitalized suicide prevention practices and, indeed, in what ways it can and should (not) happen.

**Keywords:** suicide prevention; healthcare; digitalization; subjectivity; law; ethics

## 1. Introduction

In a range of western countries over the last decades, suicide rates have been either not going down or, in some cases, increasing [1,2]. Certain groups, such as middle-aged men, teenagers and adolescents, and LGBT-I populations are disproportionately affected [3–6]. Traditional approaches to suicide prevention have not been as successful as hoped for [7,8]; and attempted suicide is notoriously difficult to predict, even for those who work with suicidal patients on a regular basis [9,10]. A range of public and private actors, including social media companies, have started to use big data and Artificial Intelligence (AI) to construct risk scores that may help predict who is most at risk of committing suicide, or use mobile health technologies to enable contact with people who are suicidal [7,8,11–16].

The following examples will serve to set the scene. First, at the end of 2018, Facebook revealed it had started to use algorithms in the US to examine who of their users might be at heightened risk of suicide, with the police sent for a 'wellness check' if the risk is deemed serious by human moderators. This program is not used in the EU because of the General Data Protection Regulation (GDPR) [11]. In 2014, the UK charity Samaritans used Twitter data for a similar prevention program, which was shut down within two weeks because of legal and privacy concerns [13]. In addition, the Dutch not-for-profit organization 113 Suicide Prevention has an e-health module that people can follow, and the organization also offers online support from therapists, which thus increases the ways in which people can contact volunteers and therapists.

However, these digitalized approaches to suicide prevention (referred to here as digitalized suicide prevention) change relationships between the involved actors, and they raise ethical, social and

legal issues for all involved, be they mental health professionals, social media companies, researchers, people experiencing suicidal thoughts, and their families and friends [11,14]. For instance, while such digital technologies may help clients deal with crises, they may also lead to difficulty establishing boundaries, such as in relation to when professionals are (not) available [17]. Hence, the use of these different technologies for suicide prevention can lead to additional challenges, with questions of privacy, boundaries, and professionals' duty to report potential harm making this into a potentially ethical minefield [18].

Moreover, it raises questions about what 'care' is, as private companies' initiatives to prevent suicide might be considered care practices even though they are not normally seen as 'care'. While most guidelines for psychotherapists and other health professionals are at a national level, one of the characteristics of contemporary initiatives for suicide prevention is that they are often adopted simultaneously in a range of countries, by actors not traditionally associated with (mental) health care. The initiatives then interact with different national legal and professional approaches, since how suicide is regarded and responded to depends partly on cultural attitudes [19–25].

While digitalized (mental) health more broadly has received scholarly attention across disciplines (for some examples, see: [26–28]), studies exploring the social, ethical, and legal aspects of digitalized suicide prevention, and particularly across a range of actors and institutional practices, are scarce. Studies that have focused on some of the legal and ethical issues associated with digitalized suicide prevention do so mostly from a health perspective, rather than drawing on social scientific literature that critically addresses suicide prevention. The aim of this article, then, is to: (a) explore two key legal and social issues related to digitalized suicide prevention; (b) by drawing on psychiatric and health services literature discussing digitalized suicide prevention initiatives; and (c) by connecting also to social scientific literature, for instance that of critical suicidology and the philosophy of psychiatry, which has not generally been drawn upon in thinking through digitalized suicide prevention (although, for an exception, see: [13]). As the use of digitalized methods for suicide prediction and prevention is likely to increase in the coming years [7], it is imperative to discuss at an early stage what the ethical, legal, and social consequences might be, and, subsequently, to follow such initiatives empirically.

In this article, then, I will draw on key literature to explore what issues (might) arise in relation to digitalized suicide prevention practices. While digitalized suicide prevention can be defined in different ways, here I will only focus on *intentional* suicide prevention initiatives. Situations where therapists happen to come across information about suicidal ideation of their clients online for instance [29,30] are beyond the scope of this article. I will start with a more in-depth introduction to digitalized suicide prevention by discussing some of the initiatives that are already implemented, and address some of the issues associated with these and with potential future initiatives. Rather than addressing the breadth of issues, however, I will then zoom in on two key issues: first, the duty of care and the duty to report, and how these two legal and professional standards may change within and through digitalized suicide prevention; and secondly a more philosophical exploration of how digitalized suicide prevention may alter human subjectivity, including how it sustains a 'logic of life'. In so doing, I will engage with several strands of literature. The health literature provides valuable insight into the current state of digitalized suicide prevention, as well as some of the hopes and concerns of health professionals in relation to such initiatives. Other literature looks at the ethical and legal issues associated with one or more digitalized suicide prevention initiatives, although rarely across the spectrum of such initiatives. I will draw upon such literature, but will also adopt a more theoretical orientation from the fields of critical suicidology, philosophy of psychiatry, and the sociology of mental health and illness. These fields have not frequently explored *digitalized* suicide prevention, yet are helpful in thinking critically about suicide prevention per se.

As such, the contribution that this article aims to make is threefold: (1) to provide an in-depth reflection on digitalized suicide prevention, which has not been done in this way in the literature so far; (2) by connecting different literatures; (3) to reflect on the ways in which digitalized suicide prevention may change subjectivities in current society, and on the ways in which the health literature's focus on

the duty to report, in particular, could reinforce such subjectivities in ways that have received criticism in the literature. Thus, this article should not be read as treating two entirely different concepts and phenomena (the duty to report and processes of subjectification), but rather the duty to report is discussed first in order then to reflect on it too *in light of* processes of subjectification. But first, I will review existing digitalized suicide prevention initiatives and some of the ethical, social and legal issues associated with them.

## 2. Digitalized Suicide Prevention

Technologies are used for suicide prevention for roughly two, often interrelated, reasons: (1) better estimations of who might be at risk of committing suicide; and (2) reaching more people and in a better way. As an example of the first reason, social media data in particular is heralded as providing more and better data about people's emotional states and intentions. This has led Facebook to implement its own suicide prevention programme based on algorithmic scouring of Facebook posts. An example of the latter category would be smartphone apps to help people to manage their suicidal ideations, some of which even have an immediate contact button so that with one click a helpline or a friend is called [7]. Often these two reasons are combined in one initiative. To take the example of Facebook again, its algorithmic analysis of posts supposedly leads to better estimations of who is at risk of suicide, and the VPN data and other data from users means that those people deemed to be at high risk could be contacted–online or offline–relatively easily.

Moreover, digitalized suicide prevention is used in different 'stages' of help. Krysinski and De Leo (2017), for instance, who focus specifically on telecommunication technology for suicide prevention, distinguish between it being used for prevention ("support services for high risk populations" (p. 239), intervention (different kind of approaches to target people with suicidal ideation, either in crisis or with low-risk suicidal clients), postvention ("follow-up interventions" after a suicide attempt (p. 239), and education and training (for instance for the general population or mental health professionals). Within each stage, different technologies can produce different effects. For instance, Krysinski and De Leo give the example of the Samaritans reporting that when they are emailed, 53% of the people express direct suicidality compared to 26% of the phone calls they get. They argue this is because "e-mail communication adds an extra degree of anonymity and control, allows for sharing of emotions without having a direct witness, and gives both help seekers and counselors additional time to compose a message" (p. 242). For the purposes of this article, it is relevant to point out that some people are more likely to reveal difficult things to a chatbot than to a (mental) health professional [31]. This is important to keep in mind because it suggests that easy criticisms of the implementation of technologies in a caring context might not be the most correct or productive in thinking about the effects that certain technologies produce. Indeed, while technologies used in a care context are often described as 'cold', in contrast to the 'warm' care that health professionals can offer, this distinction has been criticized drawing on empirical research where, sometimes, people can form highly affective relationships with and through technology [32].

In thinking about the effects that specific technologies can have, and in how they should be regulated, Marks (2019) article about AI-infused suicide prevention is of particular relevance. In it, Marks makes the distinction between "medical and social suicide prevention" (p. 102). Medical suicide prevention, he argues, is based on patient records, such as those in mental health or hospitals, and usually combines multiple sources of health data to estimate an individual's risk. Medical suicide prevention is, as Marks argues, "performed by doctors, public health researchers, government agencies, hospitals, and healthcare systems" (p. 104–105). In contrast, 'social suicide prevention' draws on an analysis of behaviours outside of the health system, such as social networking, purchasing behaviour, and the use of apps. The primary example of social suicide prevention Marks gives is that of Facebook using AI to scour both public and private messages of users to estimate their risk of attempting suicide. Although there are a few examples of groups that engage in both (for instance health professionals also drawing on social media activity), this is not common, Marks states; most of

the AI-based initiatives can be classified as one *or* the other. His argument in distinguishing the two is that medical suicide prevention by definition needs to adhere to the rules, regulations and guidelines relevant for a medical setting, such as in relation to data protection, confidentiality, safety, and privacy. Social suicide prevention, on the other hand, is not bound to any such strict health guidelines, though perhaps it should be, Marks argues. While Marks suggests that some of the ethical, social and legal consequences are shared across medical and social suicide prevention, he also argues that they each may raise different consequences. For instance, sharing of data may be a particular concern in social suicide prevention initiatives, because there is no need for private companies to adhere to the Health Information Portability and Accountability Act (HIPAA).

In general, and across medical and social suicide prevention, the potential ethical, social, and legal issues associated with digitalized suicide prevention are plentiful. One of the issues mentioned in the literature, for instance, is that of stigma, especially when the data is shared with the 'wrong' people. Indeed, some authors suggest that smartphone apps and other technologies that label someone as 'high risk' might actually be "used to cyberbully individuals detected to be acutely suicidal" [10]. Relatedly, stigma and discrimination may occur when the classification of being 'at risk' comes in the hands of insurers and (potential) employers [10]. Moreover, the literature frequently points out the risk that suicide risk assessments, perhaps in particular those based on AI, produce both false positives and false negatives [11]. For people falsely identified as being at high risk of committing suicide, this may lead to unwarranted and at times rather violent interventions, for instance forced hospitalization or an unnecessary police visit, as well as to stigmatization. This is also true for some people correctly identified as being at risk of committing suicide, for whom such interventions may also be counterproductive, causing more harm than benefits [11].

Although all these issues are important to consider in discussing and implementing digitalized suicide prevention, in the next sections I will zoom in on two issues found in different strands of literature. The first relates to the duty to report, which is associated with the duty of care. The second explores how digitalized suicide prevention leads to particular processes of subjectification. I will conclude with a reflection on digitalized suicide prevention, arguing that there is an urgent need for more empirical research into the initiatives that are implemented and their consequences.

## 3. The Duty of Care and the Duty to Report

As a more general trend, also beyond suicide prevention, psychiatry has moved more and more beyond the clinic, into people's homes and daily lives over the past decades [33–35]. As such, a range of increasingly diverse actors play a (self-appointed) role in treatment and management of psychiatric illnesses, from housing organizations through to social media companies. These different actors are held to different kinds of regulation when it comes to mental health care more broadly, and suicide prevention specifically, with concepts like confidentiality, care, and privacy taking up different meanings in these diverse practices. Digitalization, then, changes the institutional framework of mental health care. These developments raise questions such as: where does care take place, how is treatment given, who plays which role (including who pays), what is (self-) care, how does it change standards, duties of care and, as a sub-set thereof, the duty to report, and, finally, who is ultimately responsible or liable for what?

The therapeutic relationship is legally and professionally protected, especially because patients in psychiatry can be seen as particularly vulnerable. Glas (2019) refers to this as "a legal 'deepening' of the professional-patient relationship [which] is expressed by different forms of legal protection of the patient and contributes to a deeper sense of one's responsibility as professional. There are aspects of this relationship that are so vulnerable and precious that they need legal protection, as an expression of public recognition of this preciousness" [36] (p. 44).

An important contextual factor to understand here is the way in which suicide and suicide prevention are framed. It is increasingly seen as a *health* problem, "particularly one that occurs in the context of a mental illness [ . . . ]. Psychiatrists and other mental health professionals are, therefore,

increasingly expected to prevent their patients from committing suicide. Failure to do so makes mental health professionals liable for malpractice litigation, and in fact, is one of the leading reasons for successful malpractice suits against mental health professionals and institutions" [23] (see also: [25]). There are, therefore, procedures that mental health professionals need to follow when they deem their client to be at risk of hurting themselves or others. When working with clients with (acute) suicidal behavior, mental health professionals have to seek a balance between confidentiality and protecting clients' best interests [23,25]. In the professional code for Dutch psychotherapists, for instance, a large section relates to confidentiality, but it also states that, where there is a 'conflict of obligations', psychotherapists may break confidentiality if they carefully follow certain steps (https://assets.psychotherapie.nl/p/229378//files/NVPdocs/beroepscode%202018.pdf). This conflict of obligations is generally about protecting people (either the client themselves or others) from harm.

Because of digitalized technologies, this duty to report is likely to be extended to different, harder to reach, settings and populations [37]. Through technologies like video-chat, therapists can have contacts with people they have never seen in real live, and do not live close to. They may even live in different jurisdictions. Because of this, some argue that "psychotherapists and other mental health professionals [are] advised against counseling and treating suicidal patients online", because of "limited risk assessment opportunities due to lack of visual clues, client's autonomy, as well as limited access to consultation, referral, emergency care, and hospitalization services" [8]. Some even suggest that email needs to be fully prohibited from use in therapeutic practices, because it is much harder to thoroughly assess clients through this medium when they present with suicidal ideation [8]. At the same time, some authors have argued that treating people online might be more effective and reach more people who might not otherwise find their way to mental health care [7,10] so that a full prohibition of such technologies for clients who may be(come) suicidal might not be the best advice. This may lead to a conflict of obligations, too, where providing care through different mediums might be helpful but might also be difficult when it comes to crisis situations (cf. [17]).

Moreover, others have argued that the strong focus on suicide prevention may not always be productive in caring for people. It may, for instance, have consequences for health professionals; indeed, some suggest that "[i]t is important to differentiate between reasonable standards of care to prevent suicide and the utopian goal of absolute suicide prevention." [25]. Along those lines, Sisti and Joffe (2018) argue that a "corollary of the zero suicide model [aiming to get to zero suicides], [...] is that every suicide represents a culpable failure on the part of health professionals" [38] (p. 1633). In addition, and echoing the argument above, it has been suggested that preventing suicide, and protecting life, is not the only obligation mental health professionals have. Indeed, as Mishna et al. (2002) state: "[i]f a person is living with intractable depression, and his or her suffering is evident and prolonged, then the professional obligation to protect and support life competes with the obligation to alleviate suffering", which might include exploring possibilities for euthanasia (p. 270) Moreover, Mishna et al. (2002) suggest that it may, "paradoxically, [...] foster hope in the client" if a therapist is able to empathically engage with a client's suicidal thoughts and ideation, rather than immediately report clients (p. 271). There is even some anecdotal evidence that, again paradoxically, supporting someone's decision to die instills hope in such people, who know they have a humane way out should this be needed, and therefore hold off, at least initially, on going ahead with physician-assisted suicide.

Such a conflict or dynamic between alleviating suffering and preventing harm by suicide is also visible in the Dutch policy and mental health context, where, on the one hand, suicide is high on the political agenda but, on the other hand, in cases of unbearable suffering that cannot be treated anymore in the opinion of psychiatrists physician-assisted suicide is allowed also for people with mental illnesses [39–41]. A recent newspaper article has suggested that psychiatrists in particular, compared to other physicians, are very reluctant to participate in euthanasia, even when they acknowledge unbearable suffering, with people with psychiatric illnesses who have a death wish more likely to seek help with the Dutch End of Life Clinic ('Levenseindekliniek'), compared to people with for instance

terminal cancer or dementia who are more likely to be helped through routes other than this 'last resort' [42].

While these are relatively old discussions, the foundations of which might not be transformed by the advance in digital technologies used for suicide prevention, digitalization does change the mediums in which clients can be contacted and might, as such, raise additional concerns or reinforce existing concerns, for instance where therapists can be more, and differently, available compared to in the past [17]. Another issue that is maybe more transformative, however, is that other actors increasingly play a role in suicide prevention while not adhering to the same standards of care (and indeed, one can wonder whether they provide care at all). Some arguments have been made to extend the duty to report but also the duty of care to other actors that do not, traditionally, have a mental health role (such as Facebook's self-appointed role in suicide prevention) [43]. While it is not my intention here to solve this issue or suggest concrete solutions, I would argue it is imperative for a range of (policy) actors to think about the suicide prevention initiatives of private companies and how to regulate these. Privacy regulations, such as the GDPR, can perhaps play a role in forestalling the spread of initiatives, and have indeed done so in the EU [11], but as we have seen privacy is not, and should not be, the only concern. More guidelines and regulation concerning the duty of care and the duty to report also for those actors not traditionally bound to health guidelines might be needed. Furthermore, the duty to report and how it changes (or not) because of digitalized suicide prevention initiatives would be highly interesting to explore both empirically (how it is done in practice) and normatively (how it is laid out in guidelines). Such empirical and normative research could help us appreciate what can and cannot be done, and what should and should not be done, in the context of digitalized suicide prevention, and could inform the regulation of digitalized suicide prevention beyond privacy concerns.

## 4. Subjectivity

A second key issue that I wish to flag here is how digitalized suicide prevention affects human subjectivity. Foucault [44–46] famously posited that subjects come into being through (discursive, material) practices of power, and he referred to this process as subjectification. Through processes of power/knowledge, individuals are constituted [44,45], which has consequences for how we think about, act upon, and judge ourselves as individuals and within society [47]. As such, discourse and interrelated technologies, in determining how subjects can be spoken about and acted upon, produce certain kinds of, for instance, suicidal subjects, and determine what can and cannot be said about suicide.

One strand of literature that is particularly relevant here in thinking through processes of subjectification through digitalized suicide prevention is that of critical suicidology. Ian Marsh (2015) has defined this as "identifying and questioning the underlying assumptions the field [suicidology] operates within; paying close attention to the context in which they have come to be formed (including relations of power); and analyzing the effects of constituting suicide in the ways we do." [48] (p. 6). Thus, suicide and suicide prevention can be thought of in different ways, and in critical suicidology authors are interested in reflecting on how these have been constituted and with what effects. In thinking about how suicide and suicide prevention are constituted, Marsh argues that the science of suicide is dominated by only a few (medical) disciplines, notably that of psychiatry and psychology, as we have seen in the section of the duty of care and the duty to report as well. The involvement of these disciplines, and the marginalizing of others like anthropology, means that suicide comes to be considered and conceptualized in particular ways, that could also have been otherwise. In particular, "suicide is constituted primarily as an issue of individual mental health, and in relation to research particular forms of knowledge generation are strongly favored over others" [48] (p. 5). Marsh refers to this as "psychocentrism": "the reducing of human problems to flaws in individuals bodies/minds", and he calls for a "post-suicidology" that would "usefully read suicide as an ethical, social and political issue, not just one of individual pathology" (p. 7). Similarly, sociologists of mental health and illness and critical suicidology alike have argued that the focus of prevention efforts is often

on individuals and their psyche, and ignore the socio-economic circumstances that may contribute to mental illness [49,50]. While we saw in the section above that suicide prevention is mainly seen as the task of (mental) health professionals, the argument here is even more encompassing, arguing that suicide is discursively constructed mainly in relation to mental health and psychopathology more specifically. This has consequences for how we can think about, act upon, and judge ourselves and others (see also [47] for a similar argument about the psy-sciences more generally)—i.e., processes of subjectification.

Similar concerns, especially about suicide explained in terms of individual pathologies, have been raised in the philosophy of psychiatry. Indeed, authors here have argued that death wishes are commonly pathologized and medicalized, with no consideration that wanting to be death may, for some people and in some situations, be a rational choice [19,22,51]. Phenomenological studies have helped to describe how for some people, in fact, it can be rational and intelligible to think about ending one's life. Van Wijngaarden et al., for instance, focus on elderly people who are not terminally ill but consider their life complete [51], and Hewitt focuses on people diagnosed with schizophrenia who feel that a life with deteriorating and isolating illness is not worth living [19]. Medicalizing such death wishes, these authors argue, does little to understand people in their social and cultural circumstances, to respect their autonomy, and ultimately to provide care that focuses on hope-giving (see also: [22,25]). Van Wijngaarden et al. (2016) argue that such "forms of medicalization entail an epistemic risk, as conceptual, epistemic transformation not only *redefines* but also *re-designates* human life" [51] (p. 268, emphasis in original), in line with the argument that ways of framing suicide and suicide prevention produce certain kinds of subjects and enable some interventions while disabling others.

Taking the Foucaultian 'critique' even further, Tack has argued that the notion of suicide prevention is situated in a 'logic of life', where to want to end one's life is deemed unnatural, and something to be fixed [52] (for a similar argument, see: [53]). She states "that the imperative of prevention in discussions of suicide presumes that the desire to live is a natural characteristic of bodies and that this presumption means that suicide prevention is positioned as the only possible response to suicide" (p. 47). The desire to life, then, "is positioned as pre- or extra discursive" (p. 48), something that is given and is not open to contestation or even, in the extreme, to being thought about. Yet, "the subject that wants to live is itself a subject that is shaped by those who claim to merely describe it" (p. 48), hence not a natural state or pre-given. Her analysis helpfully explores how the normalizing discourse around longevity renders other kinds of subjects "as unintelligible, as impossible, as pathological, and in need of correction" (p. 55).

This echoes other authors who have argued that "the value of life is so fundamental and unquestioned in our society that the very fact of an individual questioning this value is considered irrational and a sign of illness" [25] (p. 268). While such authors do not suggest, necessarily, that nothing should be done for people with suicidal thoughts, they do contend that the emphasis on prevention is normative and political, and can, or maybe should, be up for discussion, rather than accepted without question [52]. Authors like Petrov [22] have argued for "reflexive suicide prevention efforts–as opposed to technocratic and bio-political ones- mean[ing] that one tries to meet the suicidal individual openly and fully" (p. 360). He suggests that, "[p]erhaps, instead of *preventing* death, more effort could be made to create conditions of life, that is, community, communication and commitment" (p. 361, emphasis in original). Here, the arguments touch upon those in the 'duty of care and duty to report' section, where hope-giving was also seen as sometimes in conflict with a 'reflexive' duty to report. As such, the duty to report is part of the normalizing discourse around suicide and suicide prevention that leads to certain processes of subjectification, such as that of the irrational suicidal subject that needs to be protected from themselves, and where mental health professionals play a key role in preventing suicide.

Tack, moreover, argues that it is telling that popular media and scholars alike make a sharp distinction between euthanasia and suicide, where euthanasia is often seen as a good thing, preventing "good people [to die] bad deaths", as one podcast called it, yet suicide seen as something to be

prevented [52] (p. 56). Indeed, suicide is generally constructed to be 'irrational', whereas euthanasia, such as by refusing treatment or substances, seen as generally 'rational' regardless of whether a psychiatric illness is present that might cloud one's judgment and autonomous decision-making [25,54]. Illustratively, as Leeman (2009) has argued, "the recent APA Practice Guideline for the Assessment and Treatment of Patients With Suicidal Behaviors [ . . . ] does not even mention the *possibility* of rational suicide. Perhaps this is because of what has been referred to as psychiatry's "reflexive antagonism to behaviors that hasten death"" [54] (original quote from: [55]). Leeman argues that psychiatrists owe it to their patients to treat each of their situations as unique, thus not immediately dismissing every case of suicidal ideation as irrational (see also Hewitt 2010). Interestingly however, recent evidence on capacity evaluations of psychiatric patients in the Netherlands who request euthanasia points to an opposite trend, where psychiatrists presume capacity (and thus rationality) despite evidence of a mental illness present that could cloud a patient's judgment [39].

Another, related, issue is the effect that risk assessments for suicidality may have on people and in mental health settings more generally. Some authors have argued that the emphasis on risk assessments to deal with suicidal behavior in mental health settings may work counter-productive, and prevents therapists and clients from establishing genuine, helpful connections [56,57], or provide therapists with a sense of false reassurance [58]. In addition to risk assessments often not able to truly determine who is at risk, Mulder [58] also argues that "[p]atients may [ . . . ] be detained not for treatment needs but because not detaining them produces intolerable anxiety in the staff involved in the assessment" (p. 606). Such interventions can have disastrous consequences, as Marks [11] also pointed out in the context of Facebook's suicide prevention program. Hospitalization can have negative consequences for true and false positives alike, and furthermore interventions by the police may in extreme cases be mortal–which the literature refers to as 'suicide by cop', where the police checks up on someone reported as behaving erratically or with suicidal ideation and, for whatever reason, use their gun on this person [11].

These arguments together point to particular consequences of framing suicide and suicide prevention in the ways that are currently dominant, and also begin to suggest alternatives, in which, perhaps, the wish to die is not immediately seen to be irrational and where the socio-economic circumstances of people are explicitly acknowledged and taken into account when offering help. The duty to report is part of a wider normalizing discourse that constructs mental health professionals as mainly responsible for reporting harms and preventing suicide, constructs suicide mainly as part of individual pathology, and constructs death wishes as irrational. As digitalized risk assessments likely intensify and transform the emphasis on prevention and risk, it is crucially important to empirically examine what technologies of prevention and risk *do* in diverse care practices and for the different actors involved.

## 5. Conclusions

The ways in which suicide is seen and treated differs extensively across times and cultures. In the west, for instance, it has been considered a sin, a crime, and, more recently, as a "mental accident", and all of these are political choices and carry political consequences [22,24]. In this article, I have tried to take seriously the idea that digitalized technologies also re-conceptualize suicidal ideation and re-orient treatments of suicide prevention, although how exactly is to a large extent an empirical question. One of the interesting shifts happening is that, over time, suicidal ideation has become increasingly framed as a mental health issue, with mental health professions one of the key professions to address it. On the other hand, there has been an increase in private companies like social media companies claiming they have the data to better predict who is at risk of attempting suicide on any given moment. If thinking from theories of 'medicalization' [59,60], it seems that both a medicalization *and* a de-medicalizing of suicide prevention is happening, with a noticeable lack of empirical studies looking into exactly what is happening and with what consequences.

I have argued that with the plethora of new technologies for suicide prevention, a range of ethical, legal, and social issues arise or are reinforced compared to non-technological approaches to suicide prevention. In particular, I have focused on the duty to report and on processes of subjectification, to think through the potential effects of digitalized suicide prevention. In regards to the duty to report, technological advances in therapeutic practices and beyond may mean that mental health professionals are more, or in different ways, available, which suggests a need to set appropriate boundaries to protect both people with suicidal ideation and mental health professionals. Moreover, while mental health professionals have long had particular safeguards and procedures for when and how to report someone, such safeguards and procedures are mostly lacking when it comes to social media companies for instance undertaking suicide prevention. Even for health professionals themselves, some authors have suggested a complete prohibition of the use of such technologies, such as email or video chat, for clients who may be suicidal [8]. As such, it is crucial to think about how to ensure that all actors involved with preventing suicide think about which steps they can and should take when deeming someone at high risk, and how to regulate this responsibly. Such regulations may follow Marks' suggestion that 'soft' interventions in reaching out to people are to be preferred over harder interventions like 'wellness checks' by the police and forced hospitalization [11].

Furthermore, some authors have argued that this duty to report may at times conflict with a duty to alleviate suffering. Indeed, the strong emphasis on prevention and risk assessment may lead to unintended consequences, with people feeling less rather than more cared for, and with mental health professionals having a 'false sense of reassurance' [58]. This is partly because of the likelihood of both false positives and false negatives, where false positives may lead to unnecessary and stigmatizing intervention and false negatives to those at risk not properly targeted [11,58]. In this respect, it is interesting that "[c]ompleted suicide has been found to be rarer in groups in which suicidal thoughts and suicidal behavior are more common, such as in women and adolescents – in contrast to men and the elderly, where the reverse is true" [22] (see also: [58]), which suggests important limits to the value of risk assessments.

Secondly, I have focused on processes of subjectification through digitalized suicide prevention. The discourse around suicide prevention, some authors argue, sustain a 'logic of life', where death wishes cannot be understood unless, potentially, in the context of euthanasia. That is one reason why often a strict distinction is made between euthanasia and suicide, even by people who problematize this distinction themselves [52]. This has consequences for how people with suicidal thoughts, health professionals and wider society come to understand suicidal ideation, how they judge it, and how they act upon it (cf. [47]). Indeed, discursive and material practices of suicide construct it as being irrational, a sign and consequence of mental disorder, rather than a logical response to sometimes hopeless situations. Such an approach may prevent people from committing suicide, but it may at times also mean that mental health professionals and others are more focused on preventing harm than on empathically engaging with clients or, perhaps, alleviating suffering. Moreover, it may mean that socio-economic circumstances are not properly addressed in efforts to help people with suicidal ideation, and yet such circumstances are strong risk factors for attempted and completed suicide [49,50].

At the same time, the exact consequences of digitalized suicide prevention for both reporting and caring practices and for processes of subjectification are largely an empirical issue. Therefore, it is noteworthy that the use of such technologies has not generally been the subject of empirical research, whether quantitative or qualitative. As the use of digitalized methods for suicide prediction and prevention is likely to increase in the coming years [7], it is imperative to discuss more broadly what the ethical, legal, and social consequences might be, but more importantly to follow such initiatives empirically, for instance in examining the ways in which digitalized suicide prevention interacts with sociocultural and socio-economic elements. To paraphrase the by now famous adagio, the first of the so-called 'Kranzberg's Laws' [61], digitalized suicide prevention is neither good nor bad, nor is it neutral, and it is crucial for sustained academic and social conversation to take place about who can and should be involved in digitalized suicide prevention practices and, indeed, in what ways it

can and should (not) happen. Such a conversation needs to be informed by a better understanding of what actually happens in the diverse practices that constitute digitalized suicide prevention and with what consequences.

**Funding:** This research received no external funding.

**Acknowledgments:** I would like to thank my colleagues at the Department for Law, Technology, Markets, and Society, and in particular Esther Keymolen, Linnet Taylor, Ronald Leenes, and Bert-Jaap Koops, as well as Bethany Hipple Walters, for their help in developing the ideas presented here. I would also like to thank Lambèr Royakkers for providing me with the opportunity to be included in this special issue.

**Conflicts of Interest:** The author declares no conflict of interest.

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
