# Peer review of "Technology for Our Future? Exploring the Duty to Report and Processes of Subjectification Relating to Digitalized Suicide Prevention"

_information, doi:10.3390/info11030170_

Round 1

Reviewer 1 Report

  1. Lines 354 & 355: instead of "policy," did you mean "police"?
  2. Duty to Report is not always in opposition of Duty to Care - many would argue that the first is part of the second. The author’s thesis for this section is that “digitalized suicide prevention may change relationships between (sic) the different actors…” (lines 177-178), but I don’t see how this is a unique argument involving suicide prevention - as the author points out, there are many issues with digital mental health care as a whole. It’s not controversial to claim that digitalizing some service changes relationships.
  3. This article might actually be two papers - one about how digitalized suicide prevention might impact the scope of “care” people receive from various entities, and one about how subjectivity is increased with the digital shift to suicide prevention. The author does a good job framing the debates with interesting and germane literature, but one has to read carefully to see the author’s own assertions that are separate from those quoted; thus, it’s hard to respond to the author’s points, because the idea is often one from another source. Increased emphasis on the author’s normative argument would be additive, along with a stronger link between the two parts of the paper. The general scope of “ethical, legal, and social issues” and digital suicide prevention is far too broad. Overall, the paper lacks flow, and I find myself re-reading separate parts of it and trying to find the link among them all. However, the author does make interesting points and a separate paper from each section, that elaborates more of the theories behind each position, would be very interesting. I think there is an interesting argument to be made about the impact digitalized suicide prevention may have, but it should be more clearly articulated. 

Reviewer 2 Report

This article addresses a developing area of concern relative to how social media intertwines with suicidality and suicide prevention. The author makes the case that more research needs to occur at the intersection of social media and suicide and this is certainly important. Essentially, the author creates this case from literature review. Some important dimensions are addressed including a critical perspective on suicide and efforts to conceptualize what it may be. While there is some mention of how there could be a shift of focus from individualized pathology to examination of sociocultural contexts that impact suicide, it seems that this area could be highlighted even more - or at least be emphasized as an area for more research, e.g., what sociocultural elements are associated with greater risks for suicide. In any case, the article has merit in its attempt to provoke the reader to consider how suicide may be (re)conceptualized and how we may examine the relationship of suicide and social media.
